# Continuity of Care Is Associated with Medical Costs and Inpatient Days in Children with Cerebral Palsy

**DOI:** 10.3390/ijerph17082913

**Published:** 2020-04-23

**Authors:** Kuang-Tsu Yang, Chun-Hao Yin, Yao-Min Hung, Shih-Ju Huang, Ching-Chih Lee, Tsu-Jen Kuo

**Affiliations:** 1Division of Gastroenterology and Hepatology, Department of Internal Medicine, Kaohsiung Veterans General Hospital, Kaohsiung 81362, Taiwan; 2Research Center of Medical Informatics, Kaohsiung Veterans General Hospital, Kaohsiung 81362, Taiwan; 3Department of Internal Medicine, Kaohsiung Municipal United Hospital, Kaohsiung 80457, Taiwan; 4School of Medicine, National Yang-Ming University, Taipei 11221, Taiwan; 5Yuh-Ing Junior College of Health Care and Management, Kaohsiung 80776, Taiwan; 6Department of Pediatrics, Kaohsiung Veterans General Hospital, Kaohsiung 81362, Taiwan; 7Department of Otolaryngology, Head and Neck Surgery, Kaohsiung Veterans General Hospital, Kaohsiung 81362, Taiwan; 8Institute of Hospital and Health Care Administration, National Yang-Ming University, Taipei 11221, Taiwan; 9Department of Stomatology, Kaohsiung Veterans General Hospital, Kaohsiung 81362, Taiwan; 10Department of Marine Biotechnology and Resources, National Sun Yat-sen University, Kaohsiung 80424, Taiwan; 11Department of Dental Technology, Shu-Zen Junior College of Medicine and Management, Kaohsiung 82144, Taiwan

**Keywords:** continuity of care, children with cerebral palsy, medical costs, inpatient days

## Abstract

*Background*: Children with cerebral palsy (CP) place a considerable burden on medical costs and add to an increased number of inpatient days in Taiwan. Continuity of care (COC) has not been investigated in this population thus far. *Materials and Methods*: We designed a retrospective population-based cohort study using Taiwan’s National Health Insurance Research Database. Patients aged 0 to 18 years with CP catastrophic illness certificates were enrolled. We investigated the association of COC index (COCI) with medical costs and inpatient days. We also investigated the possible clinical characteristics affecting the outcome. *Results*: Over five years, children with CP with low COCI levels had higher medical costs and more inpatient days than did those with high COCI levels. Younger age at CP diagnosis, more inpatient visits one year before obtaining a catastrophic illness certificate, pneumonia, and nasogastric tube use increased medical expenses and length of hospital stay. *Conclusions*: Improving COC reduces medical costs and the number of inpatient days in children with CP. Certain characteristics also influence these outcomes.

## 1. Introduction 

Cerebral palsy (CP), the most common childhood-onset and permanent physical disability in numerous countries, is characterized by a neurodevelopmental disorder that affects muscle tone, movement, and motor skills. By international consensus in 2005, CP describes a group of permanent disorders of the development of movement and posture causing activity limitation, that are attributed to non-progressive disturbances that occurred in the developing fetal or infant brain. CP affects approximately 1 in 500 newborns with an approximate prevalence of 17 million people worldwide [1,2,3]. We applied the Gross Motor Function Classification System (GMFCS) as the gold standard to classify motor function in children with cerebral palsy. The levels of GMFCS have been described as follows: I: Children perform gross motor skills such as running and jumping, but speed, balance and coordination are limited; II: Children have only minimal ability to perform gross motor skills such as running and jumping; III: Children use wheeled mobility when travelling long distances and may self-propel for shorter distances; IV: children are transported in a manual wheelchair or use powered mobility; V: Children are limited in their ability to maintain antigravity head and trunk postures and control leg and arm movements [2].

The average lifetime economic cost per patient with CP assessed by the US Centers for Disease Control (CDC) was US$921,000 in 2003 [4]. A Dutch study reported that the annual cost of severe CP was €40,265 (US$50,855) per child [5]. In South Korea, the total cost of rehabilitation treatment per patient with CP aged 0 to 6 and 7 to 18 years was ₩8,066,013 (US$7259) and ₩7,448,412 (US$6704), respectively [6]. A Danish report stated a lifetime CP cost of approximately €860,000 (US$971,800) and €800,000 (US$904,000) in men and women, respectively—which are higher than the US CDC estimate [7]. Moreover, these costs may be a minimum value because certain costs would not have been extracted from the database. Patients with CP are a large burden on hospitalization resources. Toyokawa et al. reported 20.6% of patients with CP aged <20 years had been admitted in 1 year (over 2012–2013) [8]. In Australia, Meehan, et al. reported that 53% of patients with CP had at least one same-day admission, and 46% had one or more multiday admissions from 2008 to 2012 [9]. Furthermore, patients with concurrent epilepsy or moderate to severe motor impairments (GMFCS III-V) had even more admissions. Primary and secondary prevention measures may reduce unnecessary and resource-wasting hospital admissions.

“Doctor-shopping”, implicating frequent attendances and switching of physicians, has become common because of easy accessibility to local clinics or hospitals in Taiwan [10]. Patients can select any physician freely. Taiwan does not possess a well-established referral system, leading to a large amount of waste with regard to medical expenditure and caregiving workforce. This phenomenon could lead to a discontinuity of care, which will reduce treatment quality and exacerbate disease status. Methods for alleviating the medical costs and resource consumption of CP need to be reappraised and re-evaluated.

Continuity of care (COC) comprises two core elements: care of an individual patient and care over time. It comprises three types of care: informational continuity, management continuity, and relational continuity. By applying COC, a trusting and responsible therapeutic association can be established between the patient and physician [11,12,13]. COC involves a continuous caring relationship between doctors and patients. COC has also been described as a “seamless service [14,15]”. Moreover, higher COC is related to a stronger sense of satisfaction, higher quality of life, better mental health, and lower costs [16,17].

The effect of COC on children with CP has not been investigated. Therefore, we designed a retrospective population-based cohort study to investigate the association of COC, inpatient days, and medical costs among children with CP in Taiwan. Furthermore, we surveyed the possible characteristics relevant to these areas.

## 2. Materials and Methods 

### 2.1. Study Design and Data Extraction

We retrospectively extracted clinical data from the Taiwan National Health Insurance (NHI) Research Database (NHIRD), a nationwide population-based administrative database of NHI, the beneficiaries of which include approximately 99.9% of the population (i.e., approximately 23 million residents) of Taiwan [18]. NHIRD includes registration files on demographic data, medical visits, physicians, inpatient days, medical costs, procedure codes, and diagnostic codes according to the International Classification of Diseases, Ninth Revision, Clinical Modification (ICD-9-CM). NHIRD validation studies have reported high accuracy of its data [19,20]. A study also reported real-world evidence from the Catastrophic Illness Patient Database using NHIRD data analysis [21,22].

The current study was approved by the Institutional Review Board of Kaohsiung Veterans General Hospital, Kaohsiung, Taiwan (VGHKS15-EM10-02). The board also waived the need for written informed consent because all personal identifying information used had been de-identified. Figure 1 shows the inclusion and exclusion criteria. In total, 4496 children with CP (age at diagnosis 0–18 years) were identified between 1 January 2000, and 31 December 2012, based on ICD-9-CM code 343.X and catastrophic illness certificates. Patients were followed until 31 December 2013. The index date was set as the day the patients obtained the catastrophic illness certificate, confirmed by specialists in relevant fields. Medical service expenditures exhibited minor changes over the years. An exchange rate of US$1 = NT$30 was used in our study—close to the current exchange rate.

### 2.2. Patient Demographics and Characteristics

Outpatient records were traced from the index date. Data were collected regarding outpatient visits one year after the index date, different physicians one year after the index date, age on the index date, gender, inpatient days over five years after the index date, medical costs over five years after the index date, inpatient visits one year before the index date, residential area, urbanization level, hospital level where children with CP obtained the catastrophic illness certificates, comorbidities, and mortality. The residential areas were classified as Northern, Central, and Southern-Eastern Taiwan. Urbanization level was dichotomized into urban and nonurban. Hospital levels, where children with CP obtained their catastrophic illness certificates, were categorized as medical centers and regional or district hospitals. The comorbidities considered in this research were asthma (ICD-9-CM 493), preterm labor (PTL) and small for gestational age (SGA; ICD-9-CM 765–765.19), perinatal complications (ICD-9-CM 760–764, 766–779, and V137), epilepsy (ICD-9-CM 345), pneumonia (ICD-9-CM 480–486, and 507.0–507.8), gastroesophageal reflux disease (GERD; ICD-9-CM 530.85, 530.11, and 530.81), and nasogastric tube use (ICD-9-CM 47017C and 47018C).

### 2.3. Study Variables

We compared total medical costs and inpatient days with a different COC index (COCI) and other characteristics.

### 2.4. COCI Application

We applied the COCI developed by Bice and Boxerman to evaluate the COC status [23]. Several studies have used this method for COC investigation, extracting data from health care claim databases [24,25,26,27,28]. By using the COCI method, we could acquire the distribution of CP outpatient visits to different physicians, and visit times to each physician from the index date. According to Bice and Boxerman [23], the COCI measures a patient’s COC based on the total number of visits and physicians as follows:$$COCI=\frac{\sum_{i=1}^{k}n_{i}^{2}-T}{T(T-1)},$$
where *T* is the total number of CP outpatient visits (CP being the principal diagnosis), *n_i_* is the number of times the patient visited a pediatric specialist, and *k* is the total number of pediatric specialists visited. In our study, the one-year COCI was applied to compare the medical costs and the inpatient days of children with CP over five years.

### 2.5. Statistical Analysis

The outcome variables of interest were medical costs and inpatient days over five years from the index date. Medical costs involved fees paid by both caregivers of patients and the NHI Bureau in Taiwan. The distribution of variables was described by the chi-squared test. The continuous variables were compared using a one-way analysis of variance. The relationship between medical costs and inpatient days, as well as other variables, was analyzed using multiple linear regression and residual analysis. Multiple linear regression was not appropriate because of unequal variance assumptions (Figure 2 and Figure 3). The effect of each variable or characteristic on medical costs and inpatient days was assessed by the hierarchical generalized linear model (HGLM) using a hospital-level random-intercept model [29,30,31]. The intraclass correlation (ICC) was also calculated based on estimations on the null model [32]. A 2-tailed *p* value of < 0.05 was considered significant. All statistical analyses were performed using SPSS (version 20; SPSS Inc., Chicago, IL, USA) and SAS (version 9.4; SAS Institute, Inc., Cary, NC, USA).

## 3. Results 

As illustrated in Figure 1, 3234 children with CP meeting our inclusion and exclusion criteria were identified from NHIRD. They were divided into three approximately equal groups according to patient numbers, the low (1076 patients, COCI < 0.235), intermediate (1077 patients, 0.235 ≤ COCI < 0.436), or high group (1081 patients, 0.436 ≤ COCI). Table 1 summarizes basic demographic data, clinical characteristics, medical costs, inpatient days, comorbidities, and mortality. The top three reasons for admissions were pneumonia, other respiratory conditions except pneumonia and epilepsy (excluding ICD-9-CM codes of 343). Detailed ICD-9-CM codes were shown in Appendix A. The estimated ICC was 0.980, indicating a significant clustering effect because the value was >0.059 [32].

### 3.1. Univariate and Multivariate Analysis for Medical Costs

The hospital-level random-intercept model HGLM, displayed in Table 2A,B, revealed that the average five-year medical costs for a child with CP in the low group were higher than for a child in the high COCI group in both univariate and multivariate analysis (US$1656 more in multivariate analysis, *p* = 0.016). The other characteristics associated with increased medical costs in both univariate and multivariate analysis were younger age, more inpatient visits, medical center, pneumonia, and nasogastric tube use. We performed a stratified analysis for medical costs after adjusting characteristics (Table 3A,B). The results indicated that the cost of five-year inpatient medical care was significantly higher in children with CP in the low group than in the high group (US$1660 more in multivariate analysis, *p* = 0.002).

### 3.2. Univariate and Multivariate Analysis for Inpatient Days

The hospital-level random-intercept model HGLM, displayed in Table 4A,B, revealed that the average five-year inpatient days for a child with CP in the low group were longer than for a child in the high COCI group in both univariate and multivariate analysis (8 days more in multivariate analysis, *p* < 0.001). The other characteristics associated with increased inpatient days in both univariate and multivariate analysis were younger age, more inpatient visits, living in Southern and Eastern Taiwan, no PTL or SGA, pneumonia, and nasogastric tube use. 

## 4. Discussion

This study is the first large scale population-based cohort study on COC application among children with CP using NHIRD. We demonstrated that a low COC was associated with an increased number of inpatient days and more medical costs among children with CP. Furthermore, outpatient medical costs were higher than inpatient.

No studies had been conducted on COCI among children with CP regarding medical costs and inpatient days. Directly comparing the cost-effectiveness of different treatments and interventions is challenging. A systematic review conducted by Shin, et al. reported that the diversity of the decision contexts and health care systems complicated examining cost-effectiveness. Prevention is a critical strategy for cost-effectiveness [33]. A study by Ranken Jordan Pediatric Bridge Hospital demonstrated that intensive care coordination, psychosocial therapy, family and caregiver empowerment, and transitional care might alleviate costs and unnecessary hospital stays among children with CP [34].

Currently, COCI related research has demonstrated that high COCI could reduce costs and the number of inpatient days. In Korea, a nationwide cohort study was conducted on 47,433 patients with new diagnoses of hypertension, diabetes, hypercholesterolemia, and complications of these diseases in 2003 and 2004. The results revealed that high COCI was correlated with fewer inpatient days and reduced inpatient and outpatient health care costs [35]. A study conducted by Chen et al. reported that higher COCI was associated with fewer hospitalizations and emergency department visits for patients with diabetes-related diseases. Patients with high COCI scores saved more on health care expenses than those with low COCI scores [36].

In the current study, lower COCI was associated with higher medical costs, particularly in ambulatory patients with CP. These findings indicated that changing physicians was correlated with a large increase in medical costs associated with outpatient visits, and, to a lesser extent, increased medical costs resulting from inpatient visits. The five–year emergency department visits after the index date of children with CP are shown in Appendix A. One-year COCI did not affect five-year emergency department visits significantly. In addition, firstly, our outcome of medical costs includes outpatient, inpatient, and emergency department visits totally. Second, in Taiwan, the medical costs of emergency visits are much less than in most other countries worldwide. Therefore, we did not take the emergency department visits into account in our results. In many diseases, such as acute myocardial infarction, gastrointestinal bleeding, acute kidney injury, and respiratory distress syndrome, outpatient medical costs are often lower than inpatient medical costs [37,38,39,40]. Our stratified analysis revealed that the low COCI group had more inpatient medical costs than the high COCI group. These findings indicated that children with CP and low COCI might have poorer care quality. Therefore, increased medical expenditure on treatments and recovery would be required once they are admitted. The low COCI group also had an increased number of inpatient days than the high COCI group. This result indicated that changing doctors might influence treatment quality, which would prolong the duration of treatment required during hospitalization. In our study, low COCI led to medical resource waste and an increased number of inpatient days. Higher COCI is necessary to improve care quality and economic efficiency of the health care system.

In our multivariate analysis of the characteristics of children with CP, younger age was associated with increased medical expenditure, which is similar to a study performed by Park et al. They reported that annual health care cost per person with CP decreased gradually with growing age before the age of 18 years [41]. Our findings also indicated that younger age was related to an increased number of inpatient days. However, Chiang reported that the mean lengths of hospital stay increased with growing age (9.8, 10.4, and 12.8 days in patients aged 4–12.9, 13–17.9, and 18–32.9 years, respectively) [42]. Increased inpatient visits and obtaining the catastrophic illness certificate at medical centers are objectively correlated with higher medical costs because children with CP would be brought to the same medical center several times for outpatient or inpatient visits. They could then obtain the catastrophic illness certificate, diagnosed by specialists. An increased number of inpatient visits also correlated with an increased number of inpatient days, because patients with CP who had more inpatient visits might have more serious diseases or complications. Our study revealed fewer inpatient days among children with CP living in Northern and Central Taiwan than among those living in Southern and Eastern Taiwan. According to the study conducted by Li et al., the highest resource supplementation was in Northern Taiwan, followed by Central, Southern, and Eastern Taiwan [43]. The uneven distribution of medical resources suggests that children with CP living in Southern or Eastern areas would have more inpatient days because of an inadequate workforce and medical support. In children with CP, respiratory diseases, including pneumonia episodes, are a principal cause of premature death and hospitalization [44]. Procedures, medicines, and the care-providing source involved in inpatient hospital stays lead to large medical expenses. Children with CP, as well as attacks of pneumonia, would also have an increased number of inpatient days. This phenomenon accords with research by Meehan, et al., which reported that respiratory diseases accounted for a larger proportion of children with a higher level of CP severity (GMFCS III-V) than those with a lower level of severity (GMFCS I-II) and were associated with the number of inpatient days [9]. Notably, patients with CP without PTL and SGA in our study had a higher number of inpatient days than those with PTL and SGA. These findings were unexpected because a study had reported that babies who were small for gestation age were more likely to have prolonged hospital stays [45]. We hypothesized that children with CP and PTL or SGA had a higher mortality rate in early life. Therefore, the number of inpatient days might be reduced in the subsequent follow-up. Nasogastric tube use increased expenses and inpatient days. A study on the costs of gastrostomy (G)/gastrojejunostomy (GJ)-tube feeding in North America reported that the cost of care for a patient with CP needing a G/GJ-tube was on average US$37,232, nearly 2.5-fold the cost of care without a G/GJ tube (US$15,004) [46]. Several limitations of nasogastric tube feeding for long term use have been reported, including recurrent pulmonary aspiration [47]. Therefore, nasogastric tube use may be related to an increased number of inpatient days. The results from prior studies were similar to ours.

There are some limitations to our study. First, the study was retrospective, and some confounding factors could not be extracted from the NHIRD, including the unrecorded costs of medication and treatments at patients’ own expense. Second, CP severity is often categorized using the Gross Motor Function Classification System. However, we could not extract the degree of CP severity from NHIRD. Tonmukayakul’s systematic review reported a strong relationship between CP severity and expenditure [48]. This limitation might reduce the comprehensive evaluation of medical costs on patients with CP. Third, we could not identify disease severity every time patients with CP were brought to doctors’ clinics. Visits to clinics could have also been for mild conditions, such as the common cold or mild diarrhea. Caregivers may not take the patients to see the same doctor under these circumstances. This may have led to bias in COCI calculation. Fourth, the diagnosis codes may not correspond to true situations. There is a possibility that difference between admission and discharge diagnoses existed. If a patient was discharged with only one diagnosis of CP, we could not track the real admission etiology [42].

## 5. Conclusions

This population-based cohort study demonstrated that high COCI levels in children with CP decreased medical costs and inpatient days. We suggest that a child with CP should not change pediatricians frequently, in order to decrease medical costs and inpatient days. Moreover, younger age, more inpatient visits, residential area, hospital level, PTL and SGA, pneumonia, and nasogastric tube use exhibited an effect on medical costs and inpatient days. For policy-making and cost-effectiveness regarding children with CP, further long-term and high-quality clinical studies should be conducted.

## Figures and Tables

**Figure 1 ijerph-17-02913-f001:**
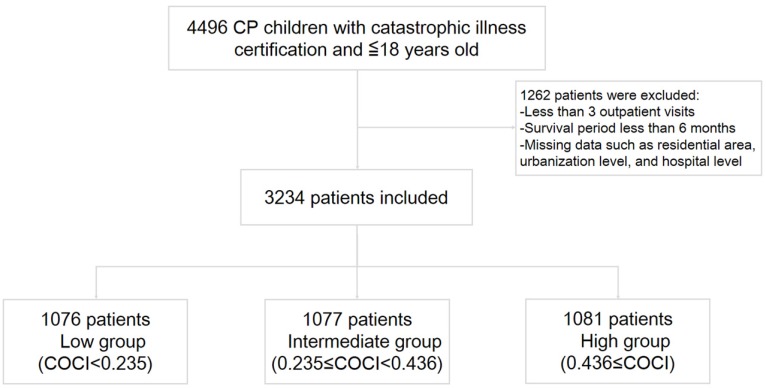
Recruitment of children diagnosed as having CP with catastrophic illness certification. Between 2000 to 2012, patients with low (n = 1076), intermediate COCI (n = 1077), and high (n = 1081) COCI. CP = cerebral palsy; COCI = continuity of care index.

**Figure 2 ijerph-17-02913-f002:**
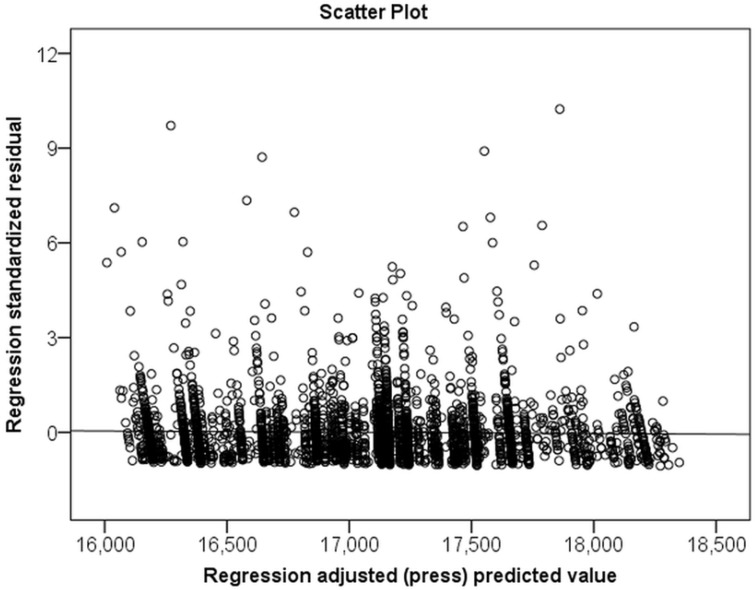
Residual plot illustrating that the standardized residual was not compatible with linear regression for medical costs over 5 years.

**Figure 3 ijerph-17-02913-f003:**
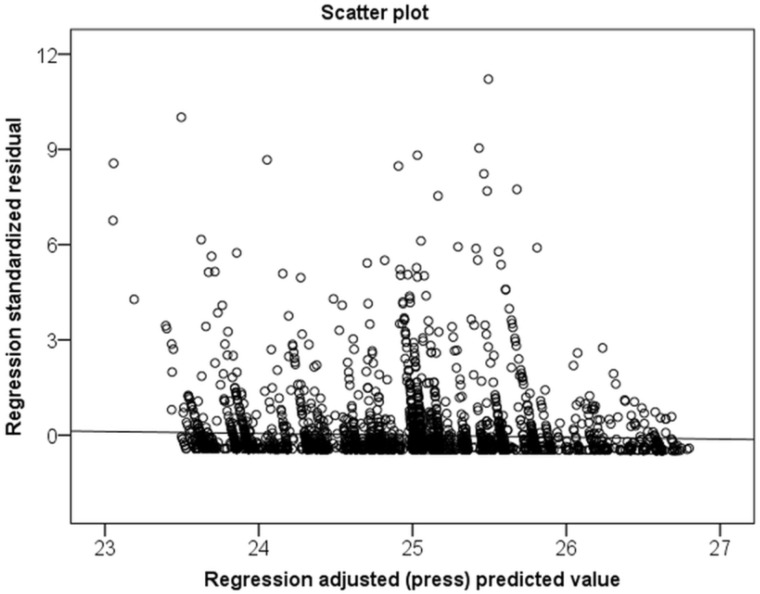
Residual plot illustrating that the standardized residual was not compatible with linear regression for numbers of inpatient days over 5 years.

**Table 1 ijerph-17-02913-t001:** One-year continuity of care index categorized according to characteristics in children with cerebral palsy.

	Total Patients	Low Group (COCI < 0.235)	Intermediate Group (0.235 ≤ COCI < 0.436)	High Group (0.436 ≤ COCI)	
Characteristics	n = 3234 (%)	n = 1076 (%)	n = 1077 (%)	n = 1081 (%)	*p*-Value
Outpatient visit					
Mean (SD)	20.1 (12.2)	19.6 (12.5)	21.8 (13.2)	18.7 (13.6)	<0.001
Medium (Q1–Q3)	17 (10–27)	17 (10–27)	19 (12–29)	15 (9–25)	<0.001
Physicians					
Mean (SD)	5.3 (3.1)	7.7 (3.3)	5.3 (2.1)	3.0 (1.7)	<0.001
Medium (Q1–Q3)	5 (3–7)	7.0 (5–10)	5.0 (4–6)	3.0 (2–4)	<0.001
Age, mean (SD), years	3.6 (3.4)	2.9 (2.8)	3.3 (3.1)	4.5 (3.9)	<0.001
Gender (Male)	1925 (59.5)	665 (61.8)	642 (59.6)	618 (57.2)	0.090
Inpatient days, mean (SD)	24.9 (55.6)	31.6 (64.5)	24.2 (52.7)	19.0 (47.4)	<0.001
Total medical costs (U.S. dollars)					
Inpatient, mean (SD)	5366 (14,062)	6836 (16,735)	4977 (12,210)	4290 (12,697)	<0.001
Outpatient, mean (SD)	11,701 (9309)	12,353 (9263)	12,091 (9266)	10,663 (9309)	<0.001
Inpatient visits, mean (SD)	1.2 (1.7)	1.3 (1.8)	1.3 (1.8)	1.0 (1.6)	<0.001
Residential area					0.028
Northern	1679 (51.9)	568 (52.8)	571 (53.0)	540 (50.0)	
Central	720 (22.3)	223 (20.7)	261 (24.2)	236 (21.8)	
Southern/Eastern	835 (25.8)	285 (26.5)	245 (22.7)	305 (28.2)	
Urbanization level					0.539
Urban	1912 (59.1)	637 (59.2)	649 (60.3)	626 (57.9)	
Non-urban	1322 (40.9)	439 (40.8)	428 (39.7)	455 (42.1)	
Hospital level					0.657
Medical center	2055 (63.5)	673 (62.5)	694 (64.4)	688 (63.6)	
Regional/District hospital	1179 (36.5)	403 (37.5)	383 (35.6)	393 (36.4)	
Comorbidity					
Asthma	450 (13.9)	136 (12.6)	162 (15.0)	152 (14.1)	0.269
PTL & SGA	1194 (36.9)	449 (41.7)	393 (36.5)	352 (32.6)	<0.001
Perinatal complication	1873 (57.9)	676 (58.4)	629 (58.4)	568 (52.5)	<0.001
Epilepsy	1638 (50.6)	509 (47.3)	572 (53.1)	557 (51.5)	0.021
Pneumonia	2051 (63.4)	732 (68.0)	699 (64.9)	620 (57.4)	<0.001
GERD	201 (6.2)	88 (8.2)	66 (6.1)	47 (4.3)	0.001
Nasogastric tube use	451 (13.9)	169 (15.7)	142 (13.2)	140 (13.0)	0.123
Mortality	568 (17.6)	203 (18.9)	192 (17.8)	173 (16.0)	0.209

Abbreviations: COCI, Continuity of Care Index; SD, standard deviation; Q1–Q3, 1st–3rd quantile; PTL & SGA, *preterm labor and small for gestational age*; GERD, gastroesophageal reflux disease.

**Table 2 ijerph-17-02913-t002:** (**A**) Medical costs over 5 years in children with cerebral palsy categorized by the level of continuity of care index and analyzed by the univariate hierarchical generalized linear model using a hospital-level random-intercept model. (**B**) Medical costs over 5 years in children with cerebral palsy categorized by the level of continuity of care index and analyzed by the multivariate hierarchical generalized linear model using a hospital-level random-intercept model.

(**A**)		
**Parameter**	**Estimated Costs (U.S. Dollars)**	***p*-Value**
COCI group		
Low group (COCI < 0.235)	3571	<0.001
Intermediate group (0.235 ≤ COCI < 0.436)	1952	0.006
High group (0.436 ≤ COCI)	Reference	
Age, year		
≤2.5	7876	<0.001
>2.5	Reference	
Gender		
Male	402	0.500
Female	Reference	
Inpatient visits		
0–1	−7828	<0.001
>1	Reference	
Residential area		
Northern	−873	0.266
Central	−673	0.477
Southern/Eastern	Reference	
Urbanization level		
Urban	−168	0.782
Non-urban	Reference	
Hospital level		
Medical center	2213	0.001
Regional/District	Reference	
Comorbidity		
Asthma		
No	129	0.879
Yes	Reference	
PTL & SGA		
No	−410	0.501
Yes	Reference	
Perinatal complication		
No	−2839	<0.001
Yes	Reference	
Epilepsy		
No	−1624	0.006
Yes	Reference	
Pneumonia		
No	−6049	<0.001
Yes	Reference	
GERD		
No	−5216	<0.001
Yes	Reference	
Nasogastric tube use		
No	−10,069	<0.001
Yes	Reference	
(**B**)		
**Parameter**	**Estimated Costs (U.S. Dollars)**	***p*-Value**
COCI group		
Low group (COCI < 0.235)	1656	0.016
Intermediate group (0.235 ≤ COCI < 0.436)	−36	0.958
High group (0.436 ≤ COCI)	Reference	
Age, year		
≤2.5	5501	<0.001
>2.5	Reference	
Gender		
Male	−137	0.809
Female	Reference	
Inpatient visits		
0–1	−3784	<0.001
>1	Reference	
Residential area		
Northern	−731	0.285
Central	−178	0.824
Southern/Eastern	Reference	
Urbanization level		
Urban	−136	0.820
Non-urban	Reference	
Hospital level		
Medical center	1173	0.044
Regional/District	Reference	
Comorbidity		
Asthma		
No	−679	0.412
Yes	Reference	
PTL & SGA		
No	380	0.600
Yes	Reference	
Perinatal complication		
No	−1187	0.088
Yes	Reference	
Epilepsy		
No	285	0.637
Yes	Reference	
Pneumonia		
No	−2513	<0.001
Yes	Reference	
GERD		
No	−1674	0.151
Yes	Reference	
Nasogastric tube use		
No	−6677	<0.001
Yes	Reference	

**Table 3 ijerph-17-02913-t003:** (**A**) Stratified analysis of inpatient medical costs for children with cerebral palsy according to the level of continuity of care index analyzed by the hierarchical generalized linear model using a hospital-level random-intercept model. (**B**) Stratified analysis of outpatient medical costs for children with cerebral palsy according to the level of continuity of care index analyzed by the hierarchical generalized linear model using a hospital-level random-intercept model.

(**A**)		
**Parameter**	**Adjusted Estimated Cost (U.S. Dollars)**	***p*-Value**
(Inpatient medical costs within 5 years)		
COCI group		
Low group (COCI < 0.235)	1660	0.002
Intermediate group (0.235 ≤ COCI < 0.436)	−458	0.397
High group (0.436 ≤ COCI)	Reference	
(**B**)		
**Parameter**	**Adjusted Estimated Cost (U.S. Dollars)**	***p*-Value**
(Outpatient medical costs within 5 years)		
COCI group		
Low group (COCI < 0.235)	614	0.121
Intermediate group (0.235 ≤ COCI < 0.436)	583	0.138
High group (0.436 ≤ COCI)	Reference	

Adjusted characteristics: age, gender, inpatient visits, residential area, urbanization level, hospital level, and comorbidities.

**Table 4 ijerph-17-02913-t004:** (**A**) Inpatient days over 5 years in children with cerebral palsy categorized by the level of continuity of care index and analyzed by the univariate hierarchical generalized linear model using a hospital-level random-intercept model. (**B**) Inpatient days over 5 years in children with cerebral palsy categorized by the level of continuity of care index and analyzed by the multivariate hierarchical generalized linear model using a hospital-level random-intercept model.

(**A**)
**Parameter**	**Estimated Days**	***p*-Value**
COCI group		
Low group (COCI < 0.235)	13	<0.001
Intermediate group(0.235 ≤ COCI < 0.436)	5	0.037
High group (0.436 ≤ COCI)	Reference	
Age, year		
≤2.5	19	<0.001
>2.5	Reference	
Gender		
Male	−1	0.974
Female	Reference	
Inpatient visits		
0–1	−39	<0.001
>1	Reference	
Residential area		
Northern	−7	0.013
Central	−9	0.016
Southern/Eastern	Reference	
Urbanization level		
Urban	−2	0.295
Non-urban	Reference	
Hospital level		
Medical center	9	0.006
Regional/District hospital	Reference	
Comorbidity		
Asthma		
No	−1	0.743
Yes	Reference	
PTL & SGA		
No	12	<0.001
Yes	Reference	
Perinatal complication		
No	1	0.913
Yes	Reference	
Epilepsy		
No	−17	<0.001
Yes	Reference	
Pneumonia		
No	−32	<0.001
Yes	Reference	
GERD		
No	−10	<0.001
Yes	Reference	
Nasogastric tube use		
No	−69	<0.001
Yes	Reference	
(**B**)
**Parameter**	**Estimated Days**	***p*-Value**
COCI group		
Low group (COCI < 0.235)	8	<0.001
Intermediate group(0.235 ≤ COCI < 0.436)	−1	0.991
High group (0.436 ≤ COCI)	Reference	
Age, year		
≤2.5	7	<0.001
>2.5	Reference	
Gender		
Male	−1	0.505
Female	Reference	
Inpatient visits		
0–1	−20	<0.001
>1	Reference	
Residential area		
Northern	−6	0.018
Central	−7	0.014
Southern/Eastern	Reference	
Urbanization level		
Urban	−1	0.528
Non-urban	Reference	
Hospital level		
Medical center	4	0.070
Regional/District hospital	Reference	
Comorbidity		
Asthma		
No	−2	0.933
Yes	Reference	
PTL & SGA		
No	6	0.004
Yes	Reference	
Perinatal complication		
No	−1	0.865
Yes	Reference	
Epilepsy		
No	−2	0.381
Yes	Reference	
Pneumonia		
No	−14	<0.001
Yes	Reference	
GERD		
No	−3	0.390
Yes	Reference	
Nasogastric tube use		
No	−54	<0.001
Yes	Reference

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
