# Peer review of "Continuity of Care Is Associated with Medical Costs and Inpatient Days in Children with Cerebral Palsy"

_ijerph, 2020, doi:10.3390/ijerph17082913_

Round 1
Reviewer 1 Report
Dear authors,
the topic of the paper is intersesting and I recommend the publication after some revisions.
P2 line 46 Please add the definition of CP
P2 line 48 please add and explain the GMFCS -levels
P2 line 56 Please give information about the severity of CP, if possible the GMFCS -level
P2 line 86 Please explain …the Catastrophic Illness Patient Database using NHIRD data analysis
P 5 Line 150 -55 recommend a statistical review
P 6 Table 1 Please give information and add the ICD Code of inpatients P8 table 4 Are there any hip or spine problems or muscle tone problems included?
P9 line 186 was any operation necessary or was performed?
P 10 line 229 Is in these areas a hip surveillance program or other programs for surveillance a standard procedure for CP patients?
P10 line 240 … children with a higher level of CP severity. Do you have any information about the GMFCS-level
P 11 line 267 Please give advises for the use and the need for surveillance program to have of the CP independent from doctors.
p2 l60: Please exchange the word "unique", it´s not only a Problem in Taiwan.
Please check the numers of the literature.
Kind regards
Author Response
Dear reviewer:
Thank you for your kindly and detailed review. We’ve provided a point-by-point revision or response to all of your comments in the following table. In the revised manuscript, all the changes are highlighted in red color. An identical manuscript without highlights has been submitted as well. We deeply apologize for the delayed reply and appreciate your valuable opinions which make this article more integrated. Thank you.
Point 1: P2 line 46 Please add the definition of CP
Response 1: Yes, the definition of CP has been added as follows (page 2, line 45-47):
By the International consensus in 2005, CP describes a group of permanent disorders of the development of movement and posture, causing activity limitation, that are attributed to non-progressive disturbances that occurred in the developing fetal or infant brain.
Point 2: P2 line 48 please add and explain the GMFCS -levels
Response 2: Yes, we have added and explained the GMFCS-levels (page 2, line 49-56):
We applied the Gross Motor Function Classification System (GMFCS) as the gold standard to classify motor function in children with cerebral palsy. The levels of GMFCS has been described as the followings, I: Children perform gross motor skills such as running and jumping, but speed, balance and coordination are limited; II: Children have only minimal ability to perform gross motor skills such as running and jumping; III: Children use wheeled mobility when travelling long distances and may self-propel for shorter distances; IV: children are transported in a manual wheelchair or use powered mobility; V: Children are limited in their ability to maintain antigravity head and trunk postures and control leg and arm movements.
Point 3: P2 line 56 Please give information about the severity of CP, if possible the GMFCS –level
Response 3: Yes, we have given information about the severity of CP with the GMFCS-level (page 2 , line 68-69):
Furthermore, patients with concurrent epilepsy or moderate to severe motor impairments (GMFCS III-V) had even more admissions.
Point 4: P2 line 86 Please explain …the Catastrophic Illness Patient Database using NHIRD data analysis
Response 4: Yes, we have explained the the Catastrophic Illness Patient Database using NHIRD data analysis. From the study [21], the breast cancer cohort was extracted from the 2 most comprehensive databases, National Health Insurance Research Database (NHIRD) and Catastrophic Illness Patient Database (CIPD), each of which covers most of the breast cancer population in Taiwan. They used the full breast cancer women cohort extracted from the NHIRD, and confirmed the cohort with the data of Registry for CIPD, for which histologic confirmation of breast cancer is required for recruitment.
Furthermore, under the National Health Insurance (NHI) program, patients with severe illnesses can apply for catastrophic illness certification, so as to be exempted from certain NHI payments and copayments for each health care encounter. All applications for catastrophic illness certification are reviewed by experts, and therefore the diagnosis can be considered highly accurate; hence, the catastrophic illness file has been used for case ascertainments in respective research [22].
Point 5: P 5 Line 150 -55 recommend a statistical review
Response 5: Yes, we have followed the recommendation. We divided the children with CP into three approximately equal parts according to patient numbers. The sentence is modified as (page 6, line 161-165): They were divided into three approximately equal parts according to patient numbers, the low (1076 patients, COCI < 0.235), intermediate (1077 patients, 0.235 ≤ COCI < 0.436), or high group (1081 patients, 0.436 ≤ COCI).
Point 6: P 6 Table 1 Please give information and add the ICD Code of inpatients
Response 6: Yes, we have given information and addded the ICD Codes in the supplementary file (page 6, line 166-168): The top three reasons of admissions were pneumonia, other respiratory conditions except pneumonia and epilepsy (excluding ICD-9-CM codes of 343). Detailed ICD-9-CM codes were shown in the supplementary file.
Point 7: P8 table 4 Are there any hip or spine problems or muscle tone problems included?
Response 7: We had tried to collect data of patients with scoliosis or receving joint, bone or hip surgery. However, the number of patients was too few. Therefore, we didn’t include the above problems.
Point 8: P9 line 186 was any operation necessary or was performed?
Response 8: We have demonstrated our results according to our table 2,3 and 4. There was no operation necessary or performed.
Point 9: P 10 line 229 Is in these areas a hip surveillance program or other programs for surveillance a standard procedure for CP patients?
Response 9: Yes, the orthopedic specialists in Taiwan certify a hip surveillance program or other programs for surveillance would be performed in outpatient clinics for children with CP.
Point 10: P10 line 240 … children with a higher level of CP severity. Do you have any information about the GMFCS-level
Response 10: Yes, we have added the information about the GMFCS-level (page 14, line 288-291): This phenomenon accords with research by Meehan, et al., which reported that respiratory diseases accounted for a larger proportion of children with a higher level of CP severity (GMFCS III-V) than those with a lower level of severity (GMFCS I-II) and were associated with the number of inpatient days [9].
Point 11: P 11 line 267 Please give advises for the use and the need for surveillance program to have of the CP independent from doctors.
Response 11: Yes, we have given advice. We have added one more sentence (page 14, line 318-319): We suggest a child with CP not change pediatricians frequently to decrease medical costs and inpatient days.
Point 12: P 2 60: Please exchange the word "unique", it´s not only a Problem in Taiwan.
Response 12: Yes, we have exchanged the word “ unique”. The sentence is modified as follows (page 2, line 71-72 ):
“Doctor-shopping,” implicating frequent attendances and switching of physicians, has become common because of easy accessibility to local clinics or hospitals in Taiwan.
Point 13: Please check the numers of the literature.
Response 13: Yes, we have checked the numbers of the literature, which is 48.

Reviewer 2 Report
Dear authors
Your work seems very important to us to visualize the impact on the costs of care in patients with sequelae of cerebral palsy in their population and the various reasons that are related to this.
It seems to us that the main problem is the format of the tables you present, since it makes their interpretation difficult because they are extensive and contain a lot of information. It is highly recommended to explore supporting your tables with some format of bar graphs to facilitate their interpretation.
Another concern is the potential bias related to the increase in the number of visits and the number of doctors (among other variables considered) with emergency events that, apparently, were not considered in the analysis. Please comment on it in the discussion.
In the body of the manuscript it is only necessary add some definitions for the non-expert reader and some modifications in the tables to facilitate the visibility and type of results, so we added it in the attached document .pdf
Greetings

Author Response
Dear reviewer:
Thank you for your kindly and detailed review. We’ve provided a point-by-point revision or response to all of your comments in the following table. In the revised manuscript, all the changes are highlighted in red color. An identical manuscript without highlights has been submitted as well. We deeply apologize for the delayed reply and appreciate your valuable opinions which make this article more integrated. Thank you.
Point 1: It is recommended to put the equivalent care costs in USD in order to easily visualize the costs in the different countries (page 2, line 49-52).
Response 1: Yes, we have put the equivalent care costs in USD. Please check the revised manuscript (page 2, line 59-62)
Point 2: Please define this phenomenon in more detail (page 2, line 60), because it is assumed that the reader knows. In most countries with social health systems institutionalized by the government and in much of Latin America it is not a common practice. It is recommended to add a simple definition for the reader unfamiliar with this practice.
Response 2: Yes, we have defined this phenomenon in more detail (page 2, line 71-72):
“Doctor-shopping,” implicating frequent attendances and switching of physicians, has become common because of easy accessibility to local clinics or hospitals in Taiwan [10].
Point 3: this table is difficult to interpret. It is recommended to separate the results of the univariate and multivariate analyzes into two separate tables and to model a matrix in a format of contingency table (cross tabulation)...
It is highly recommended to explore supporting your tables with some format of bar graphs to facilitate their interpretation.
(Same comments as table 2、3、4)
Response 3: Yes, we have revised the table format and separated them. Please check them in our revised manuscript (table 2A & 2B: page 8-9, line 191-198; table 3A & 3B: page 10, line 200-201; page 11-12, line 218-224)
Point 4: Broadening the dissemination taking into account the potential cost bias, the number of consultations and the number of physicians may be related to acute complications, which are not taken into account because they are not listed in the diseases considered in the study design. Complications during the evolution of these patients could be related to these variables evaluated and which were not considered during the interpretation of the results ...
It is recommended to take into account the number of visits to emergency services to visualize the impact of acute complications and consider them in the final discussion ...(page 9, line 207)
Response 4: Yes, we have added a supplementary table to show the emergency department visits. We have added viewpoints in the discussion (page 13, line 251-256):
The five–year emergency department visits after the index date of children with CP were shown in the supplementary table. One-year COCI did not affect the emergency departements visits siginificantly. In addition, first, our outcome of medical costs includes outpatient, inpatient, and emergency department visits totally. Second, in Taiwan, the medical costs of emergency visits are much less than most other countries worldwide. Therefore, we did not take the emergency visits into account in our results.
